# In-Vivo Somatostatin-Receptor Expression in Small Cell Lung Cancer as a Prognostic Image Biomarker and Therapeutic Target

**DOI:** 10.3390/cancers15143595

**Published:** 2023-07-13

**Authors:** Feyza Şen, Gabriel T. Sheikh, Johannes von Hinten, Andreas Schindele, Malte Kircher, Alexander Dierks, Christian H. Pfob, Sebastian E. Serfling, Andreas K. Buck, Theo Pelzer, Takahiro Higuchi, Alexander Weich, Ralph A. Bundschuh, Rudolf A. Werner, Constantin Lapa

**Affiliations:** 1Department of Nuclear Medicine, University Hospital Wuerzburg, 97080 Würzburg, Germany; 2Nuclear Medicine, Faculty of Medicine, University of Augsburg, 86154 Augsburg, Germany; 3Department of Nuclear Medicine, Pendik Training and Research Hospital, Marmara University, 34722 İstanbul, Turkey; 4Department of Nuclear Medicine, Ludwig-Maximilians-Universität München, 81377 Munich, Germany; 5Department of Internal Medicine I, Pulmonology, University Hospital Wuerzburg, 97080 Würzburg, Germany; 6Faculty of Medicine, Dentistry and Pharmaceutical Sciences, Okayama University, Okayama 700-8530, Japan; 7Department of Internal Medicine II, Gastroenterology, University Hospital Wuerzburg, 97080 Würzburg, Germany; 8Johns Hopkins School of Medicine, The Russell H Morgan Department of Radiology and Radiological Science, Division of Nuclear Medicine and Molecular Imaging, Baltimore, MD 21287, USA

**Keywords:** somatostatin receptor, SSTR, SCLC, small cell lung cancer, theranostics, DOTATATE, DOTATOC, PET

## Abstract

**Simple Summary:**

Despite novel targeted treatment options, small cell lung cancer (SCLC) still has a bad prognosis. However, as a relevant number of SCLC patients show a high expression of somatostatin receptors (SSTRs), SSTR-targeted radionuclide therapy (PRRT) may be a treatment option. Therefore, we investigated whether SSTR expression assessed in positron emission tomography (PET) has prognostic value. In patients with adequate PET uptake, PRRT was performed, and the outcome was investigated. We found that SSTR-targeted PET, although not a prognostic tool for outcome, is an important tool for treatment decision. In some patients, PRRT can be a promising treatment option as a second or third line treatment of SCLC.

**Abstract:**

Background: Given the dismal prognosis of small cell lung cancer (SCLC), novel therapeutic targets are urgently needed. We aimed to evaluate whether SSTR expression, as assessed by positron emission tomography (PET), can be applied as a prognostic image biomarker and determined subjects eligible for peptide receptor radionuclide therapy (PRRT). Methods: A total of 67 patients (26 females; age, 41–80 years) with advanced SCLC underwent SSTR-directed PET/computed tomography (somatostatin receptor imaging, SRI). SRI-avid tumor burden was quantified by maximum standardized uptake values (SUV_max_) and tumor-to-liver ratios (T/L) of the most intense SCLC lesion. Scan findings were correlated with progression-free (PFS) and overall survival (OS). In addition, subjects eligible for SSTR-directed radioligand therapy were identified, and treatment outcome and toxicity profile were recorded. Results: On a patient basis, 36/67 (53.7%) subjects presented with mainly SSTR-positive SCLC lesions (>50% lesions positive); in 10/67 patients (14.9%), all lesions were positive. The median SUV_max_ was found to be 8.5, while the median T/L was 1.12. SRI-uptake was not associated with PFS or OS, respectively (SUV_max_ vs. PFS, ρ = 0.13 with *p* = 0.30 and vs. OS, ρ = 0.00 with *p* = 0.97; T/L vs. PFS, ρ = 0.07 with *p* = 0.58 and vs. OS, ρ = −0.05 with *p* = 0.70). PRRT was performed in 14 patients. One patient succumbed to treatment-independent infectious complications immediately after PRRT. In the remaining 13 subjects, disease control was achieved in 5/13 (38.5%) with a single patient achieving a partial response (stable disease in the remainder). In the sub-group of responding patients, PFS and OS were 357 days and 480 days, respectively. Conclusions: SSTR expression as detected by SRI is not predictive of outcome in patients with advanced SCLC. However, it might serve as a therapeutic target in selected patients.

## 1. Introduction

Small cell lung cancer (SCLC) is the most aggressive subtype of neuroendocrine tumors of the lung and accounts for approximately 13% of primary cases [1]. Characterized by rapid disease progression and early relapse, even in subjects with early and guideline-compatible treatment, median survival is approximately one year, with an overall 5-year survival rate of less than 10% [2]. Despite the initial benefit of treatment, including chemotherapy (CTx) and/or external beam radiation therapy (RTx), a substantial portion of patients are prone to early relapse [1,3]. Although immunotherapy shows promising results, especially in combination with conventional chemotherapy [4], other therapy options are much needed. One potential approach that has been discussed previously is peptide receptor radionuclide therapy (PRRT). This treatment is well established in neuroendocrine tumors of the gastrointestinal tract [5] and has proven favorable results in prospective clinical trials [6]. It is based on the overexpression of somatostatin receptors (SSTR) on the tumor cell surface as a selective target for radiolabeled SSTR-analogues [7]. In SCLC, ex vivo analysis has demonstrated an upregulation of SSTR in up to 69% of cases, rendering those a suitable target for PRRT [8]. In vivo positron emission tomography (PET)/CT with Gallium-68(^68^Ga)-labeled SSTR agonists showed promising results as well. For instance, Lapa and colleagues reported high SSTR expression in sites of disease in 4 of 21 patients and an intermediate uptake in another 6 of 21 patients, still with the majority of lesions categorized as positive [9]. In this study, a good correlation between SSTR positivity in PET imaging and histopathology was noted. In another study, Sollini et al. recorded intense radiotracer accumulation in 12 of 24 SCLC patients (50%) [10]. Of note, 11 of these 12 patients also underwent PRRT with time to progression reported to be as short as 90 days, with a broad range from 7 to 238 days. However, only four patients received more than one PRRT cycle with either Lutetium-177 (^177^Lu)-labeled or Yttrium-90 (^90^Y)-labeled SSTR-agonists.

In the present study, we aimed to investigate SSTR expression in receptor-targeted PET (somatostatin receptor imaging, SRI) in a larger cohort of 67 SCLC patients, as well as the outcome and toxicity profile of PRRT in a sub-cohort of 14 subjects. We also aimed to assess the predictive value of the baseline SRI signal for progression-free (PFS) and overall survival (OS). 

## 2. Material and Methods

All subjects gave written informed consent for all diagnostic and therapeutic procedures as well as for the publication of study results. Imaging and treatment were offered according to the German Pharmaceutical Act (§13.2b) on a compassionate-use basis. The local ethical committee of Wuerzburg University waived the need for approval given the retrospective character of this study (# 20210415 03). An ad hoc analysis of this cohort with a limited number of subjects has been reported previously [9]. In the current investigation, we now present an updated cohort of our single-center experience.

### 2.1. Patients and SSTR-Directed Molecular Imaging 

A total of 67 patients (27 females, 40 males; age ranging from 41 to 80 years) with histopathologically proven advanced SCLC were enrolled. Apart from one subject, all patients had received CTx (including 1st line (cis-/carboplatin, etoposide) and/or 2nd line protocols (topotecan, gemcitabine, or ixoten)). At the time of SSTR-directed procedures, all subjects had experienced either disease progression or tumor relapse. Details on patients’ characteristics are provided in Table 1.

Images were acquired using an integrated, full-ring lutetium oxyorthosilicate PET- and 64-multislice CT-scanner (Siemens Biograph mCT 64, Siemens Healthineers, Erlangen, Germany). PET and spiral CT scans were performed 40–60 min after intravenous injection of [^68^Ga]DOTATATE (123 ± 30 MB). For further details on the imaging procedure, please refer to [9].

### 2.2. Scan Interpretation

Scans were assessed visually by two readers (JH, FS) with at least three years of experience in somatostatin receptor imaging using a Siemens syngo.via workstation (syngo.via software MM oncology workpackage version VA30, Siemens Healthineers, Erlangen, Germany). Maximum standardized uptake values (SUV_max_) of the hottest tumor lesion were recorded. Background activity was defined as the mean SUV (SUV_mean_) within a 5 cm spherical volume of interest (VOI) placed in the healthy tissue of the right liver lobe. Respective tumor-to-liver ratios (T/L) were then calculated. 

As described in [9], tumor manifestations were divided into positive lesions and defined by an uptake above the mean radiotracer accumulation in healthy liver tissue, and negative lesions were defined by a mean uptake less than that of the healthy liver. In addition, patients were assigned to three groups based on their lesion uptake. This classification included the “positive” group, in which all sites of disease were considered SSTR-positive; the “intermediate” group, in which the majority (>50%) of lesions were positive; and a “negative” group, in which the majority/all lesions were negative with no significant uptake.

### 2.3. PRRT

Patients with more than 80% of lesions with positive uptake were considered to be eligible for PRRT. This criterion was fulfilled by 18 patients. Two patients died before the start of PRRT and another two patients rejected PRRT treatment as well as any other further treatment, due to the increasing deterioration of their clinical status. The 14 remaining patients received a total of 29 cycles of PRRT in accordance with current practical guidance and after standard pre-examinations, including renal scintigraphy and blood sampling [11]. Detailed numbers of cycles for each patient can be found in Table 2. For each cycle, a median activity of 7.5 GBq (range, 6.0–8.5 GBq) of ^177^Lu-labeled SSTR-agonists ([^177^Lu]DOTATOC in 20 cases and [^177^Lu]DOTATATE in 9 cases) were intravenously administered. Per routine protocol, we assessed vital signs and the general patient condition on a regular basis after commencing treatment. We subsequently also performed routine follow-up assessments every 2 weeks, including blood tests to determine bone marrow and renal function [11].

### 2.4. Response Assessment

Treatment response was determined on a clinical or radiological basis (assessed by CT (performed every 3 months) and/or SRI scans (after every second PRRT cycle)). We applied Response Evaluation Criteria in Solid Tumors (RECIST) 1.1 and calculated PFS (defined as date of scan till date of first documented disease progression) and OS (defined as date of SRI till death).

### 2.5. Statistical Analysis

Data were analyzed using Python 3 and the SciPy library (version 1.6.2). Non-parametric tests were employed unless the Shapiro–Wilk test indicated that the data were normally distributed. The Mann–Whitney U-test was used to compare the means of SUV_max_ and T/L ratio between the positive, intermediate, and negative groups in this study. Survival analyses, including PFS and OS, were performed using Python 3 and the Lifelines library (Version 0.26.3). The Log-Rank test was used to compare survival times between groups. 

Regression analysis, using Spearman’s rho, was used to examine the correlation between OS and PFS and PET-derived quantitative parameters. Hazard ratios (HR) of death and progression along with a 95% confidence interval (95% CI) were also calculated. A *p*-value of less than 0.05 was considered to be statistically significant.

## 3. Results

### 3.1. Patients 

At the time of SRI, the most common metastatic sites were lymph nodes in 61/67 subjects (91%), skeleton in 35/67 subjects (52.2%), and liver in 33/67 subjects (49.3%). Further details can be found in Table 1. A total of 64 of the 67 (95.5%) patients died during follow-up, and three patients were lost for follow-up; thus, no date of death was available for analysis. However, the time of progression was still documented. The median PFS of the overall cohort was 70 days (range, 3–766 d), and the median OS was 99 days (range, 3–907 d), respectively. Age (≤60 vs. >60 years) and gender were not significantly associated with PFS (age, *p* = 0.94; gender, *p* = 0.39) or OS (age, *p* = 0.54; gender, *p* = 0.78). 

### 3.2. Somatostatin Receptor Imaging (STR-PET/CT)

Visually, 31/67 (46.3%) patients did not show any significant SSTR expression in sites of disease, while in the remaining 36/67 (53.7%) subjects, the majority of lesions were rated SRI positive. Noteworthy, in 10 of these 36 (27.8%) cases, all SCLC manifestations were rated SSTR positive. Lesions with substantial uptake higher than the SUV_mean_ of the liver in >80% of lesions could be found in an additional 8/36 (22.2%) patients. As such, from the entire cohort of 67 patients, 18 (26.8%) qualified for PRRT based on SSTR expression in SRI.

In the quantitative analysis, the median SUV_max_ of the respective hottest lesion was 8.5 (range, 2.5–115.7), and the corresponding median T/L was 1.12 (range, 0.4–23.8). Patients with just positive lesions had a median SUV_max_ of 22.7 (range, 7.3–39.1) and a median T/L of 2.9 (range, 1.3–5.8), respectively. In the “intermediate” group, the median SUV_max_ of 11.3 (range, 4.0–115.7) and the median T/L of 1.7 (range, 1.1–23.8) were not significantly lower when compared to subjects with only PET-avid lesions (*p* = 0.11 and *p* = 0.17). Individuals rated SSTR-negative (median SUV_max_, 5.0 (range, 2.5–10.6) and T/L, 0.71 (range, 0.35–1.03), respectively) exhibited significantly lower values when compared to patients allocated to the “positive” (*p* = 0.007) or “intermediate” group (*p* < 0.005, respectively). Neither SUV_max_ nor T/L were significantly correlated with outcome (SUV_max_ vs. PFS, ρ = 0.13 with *p* = 0.30 and vs. OS, ρ = 0.00 with *p* = 0.97; T/L vs. PFS, ρ = 0.07 with *p*= 0.58 and vs. OS, ρ = −0.05 with *p* = 0.70) (Figure 1).

### 3.3. PRRT 

In those 18 subjects eligible for PRRT, patients showed a mean SUV_max_ of 29.9 ± 14.6 and T/L of 4.9 ± 2.4. Out of the patients eligible for radionuclide therapy, four subjects did not receive PRRT due to premature death (n = 2) or due to increasing deterioration of clinical status and patient’s decision not to perform any further treatment (i.e., best-supportive care; n = 2). Thus, a total of 14 patients of the entire cohort were treated with PRRT.

In total, 1–6 cycles of PRRT were administered (median of 1, number of treatment cycles per patient defined by overall performance status and treatment response). Out of 14 patients, 8/14 (57.1%) received one cycle, 2/14 (14.2%) received two and three cycles, respectively, and one patient each (7.1%) received five and six cycles, respectively. More details about the treatment and the patient cohort can be found in Table 2. 

During PRRT, no acute adverse event of grade 3 or higher occurred. Treatment was well tolerated in all patients, without relevant changes in vital signs. In addition, no therapy related long-term toxicity, including kidney failure or myelodysplastic syndrome, was observed during follow-up.

Of the 14 treated patients, one subject succumbed to treatment-independent infectious complications immediately after PRRT, leaving 13 individuals for further outcome analysis. Disease control was achieved in 5/13 (38.5%), with partial remission in 1/13 (7.7%) and stable disease in 4/13 (30.8%) subjects, respectively. One patient received only one cycle prior to death due to respiratory exhaustion associated with PRRT-unrelated pneumonia prior to first SSTR-directed PET/CT restaging. Nonetheless, the last CT scan directly prior to demise revealed unchanged findings of the tumor burden. The remaining 8/13 (61.5%) experienced progressive disease (PD). Median PFS in patients treated with PRRT was 108.5 days (range, 17–766 d), and median OS was 126 days (range, 17–907 d). Subjects with controlled disease had a significantly prolonged PFS and OS with 357 (range, 115–766 d) and 480 days (range, 118–907 d), respectively, compared to the PD group (*p* < 0.005 and *p* < 0.01, respectively; Figure 2 and Figure 3).

Following progression after PRRT, chemotherapy or radiation therapy alone or a combination of both was added in 1/14 (7.1%), respectively, while in 10/14 (71.4%) subjects, no further treatment was performed, just best supportive care.

## 4. Discussion

Even today, the therapy of advanced small cell lung cancer remains challenging due to its aggressive growth behavior. Current guidelines recommend the use of CTx, thoracic radiotherapy, and prophylactic cranial irradiation [2]. Although SCLC is sensitive to chemotherapeutic protocols when combined with the checkpoint inhibitor atezolizumab as first-line treatment [12], most patients experience relapse within one year [2,12]. Several immunotherapeutic drugs have entered the clinical arena in the last decade but have yielded only mixed results, e.g., for nivolumab-ipilimumab, for which a randomized phase II study did not reach its primary endpoint of PFS improvement [13]. As second-line treatment, the topoisomerase inhibitor topotecan also demonstrated only limited benefit in platinum-resistant relapse [2,14]. There is still no established last/third-line treatment in end-stage SCLC to date. 

This report represents an update of our previous work [9], evaluating extensively pretreated patients affected with advanced-stage SCLC, with an extended number of 67 scanned patients. In patients undergoing SRI, we revealed substantial in vivo target expression (>80% of lesions positive) in 18/67 (26.9%) of subjects, who were found eligible for PRRT. Of those patients, 14 received SSTR-targeted, radiolabeled agonists, which were well tolerated. Of note, this cohort of 14 subjects receiving PRRT is the largest reported to date, which all have received homogenous treatment using ^177^Lu-labeled SSTR-directed therapy.

Among all patients treated with PRRT, four patients showed stable disease, and one patient even achieved partial response with a median OS of 16 months. This was longer than in previously reported cohorts with median survival times of 9.4–12.8 months in a relapsed or refractory setting [15]. However, in the total treated cohort, mean OS was 4.1 months with a wide range from half a month up to two and a half years, indicating that selected patients may benefit from PRRT and also experience prolonged survival benefits.

Beyond determining SSTR expression to allow the identification of patients eligible for PRRT in a theranostic third line setting, we aimed to assess a potential association between tumor heterogeneity and clinical outcome. Based on the assumption that sustained SSTR expression in all tumor sites reflects (to a certain degree) preserved differentiation and concomitant lower tumor aggressiveness, we wanted to determine whether positive patients (defined as SSTR expression in all lesions) might experience an improved outcome when compared to patients revealing none or only limited radiotracer accumulation (defined as SSTR-negative) in sites of disease. In this regard, preserved SSTR expression was not related to a superior outcome relative to those individuals with overall absent receptor expression. Not surprisingly, SUV_max_ and T/L were also not significantly correlated with OS or PFS. 

To further achieve more long-lasting therapeutic efficacy, the currently applied PRRT protocol, which followed the current practical guidance [11], may be further optimized. For instance, the time frames between consecutive treatment cycles may be reduced, e.g., from an 8- or 12- to a 4-week interval. Another optimization could be the adjustment of the applied amount of activity. In 11 end-stage SCLC patients receiving 2.6 to 6.0 GBq of ^177^Lu- or ^90^Y-labeled compounds, Sollini et al. did not achieve an objective response [10], whereas in the present analysis, a substantial proportion of treated patients showed disease stabilization after receiving a cumulative activity of at least 6 GBq of [^177^Lu]-DOTATATE or [^177^Lu]-DOTATOC. Further optimization of dose-response relation may be achieved by pretherapeutic dosimetry as a tailored treatment approach to determine optimal activity, further prolonging relevant clinical endpoints by allowing the administration of higher amounts of activity at each individual cycle [16]. Further refinement may also include the use of more potent SSTR-targeted radiolabeled antagonists or the use of targeted alpha therapy in the context of SSTR-expressing disease [17,18].

Another point to be discussed is the combination of PRRT with systemic treatments. Future studies should define the potential benefit of such combined treatment approaches, e.g., by the concomitant use of poly-ADP ribose polymerase (PARP) inhibitors or immunotherapies, such as the checkpoint inhibitor atezolizumab in addition to PRRT [2,12,19]. In this regard, a recent study reported on the feasibility of SSTR-directed radiolabeled analogues combined with nivolumab in nine patients with advanced disease. With 3/9 (33.3%) patients achieving disease control, this treatment scenario may warrant further investigation, preferably on a larger scale including more subjects [20]. 

In addition, our theranostic approach was exclusively offered as last-line treatment. Patients with PET radiotracer uptake in lesions above that of healthy liver tissue were treated on a compassionate-use basis. In this regard, future investigations should also incorporate PRRT earlier in the treatment algorithm and disease course. Nonetheless, acute adverse events were rather negligible for PRRT, and thus, given the high toxicity profile seen with other drugs for SCLC treatment [21], PRRT may still be a suitable option in those advanced cases.

## 5. Conclusions

In our extended cohort of patients with advanced SCLC, SSTR expression had no prognostic value for PFS or OS. Given the acceptable safety profile, SSTR-directed endoradiotherapy may still be a therapeutic option in patients with sufficient receptor expression, present in about a quarter of advanced-stage SCLC patients. Further optimization of PRRT in terms of shorter therapy intervals, the application of personalized dosimetry, the use of new radiopharmaceuticals such as SSTR-antagonists and alpha emitters, or in combination with already established treatment regimens (including chemo- and immunotherapy) are warranted.

## Figures and Tables

**Figure 1 cancers-15-03595-f001:**
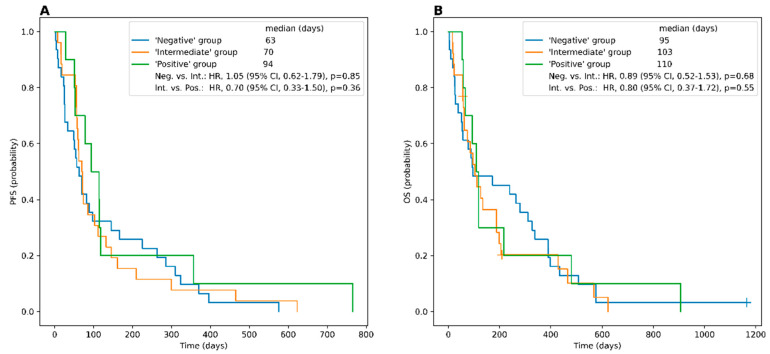
Kaplan–Meier curves and the log-rank comparison for progression-free survival (PFS) and overall survival (OS) between the different PET groups. PFS (**A**) and OS (**B**) of patients with all lesions positive (“positive” group), as compared to subjects with the majority of lesions positive (“intermediate” group), as well as the comparison between the patients of the “intermediate” group and the patients with liver uptake higher than lesion uptake in all or the majority of SCLC lesions (“negative” group).

**Figure 2 cancers-15-03595-f002:**
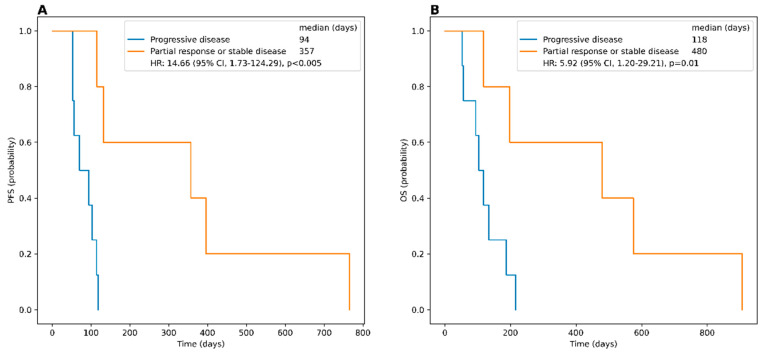
Kaplan–Meier curves and the log-rank comparison for progression-free survival (PFS) and overall survival (OS) between patients with controlled disease (partial response (PR), stable disease (SD)) and individuals with progressive disease (PD) treated with PRRT. Patients with PR or SD show significantly prolonged PFS (**A**) and OS (**B**) compared to patients with PD.

**Figure 3 cancers-15-03595-f003:**
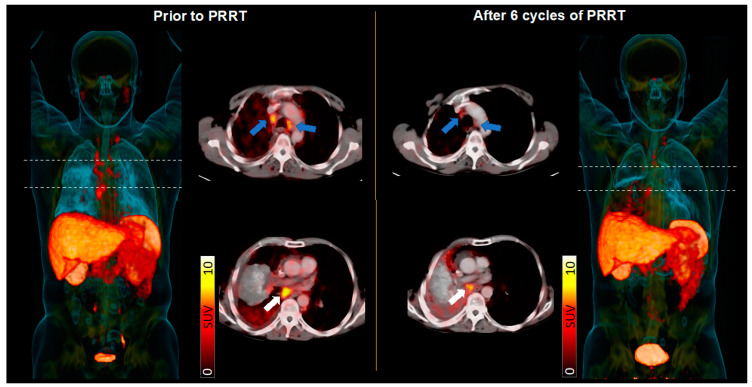
Patient with advanced small cell lung carcinoma scheduled for somatostatin receptor (SSTR)-targeted peptide receptor radionuclide therapy (PRRT). (Left): Maximum-intensity projection (MIP) and trans-axial SSTR-PET/CT at baseline revealed extensive SSTR-avid tumor burden in mediastinal (left, blue arrows) and infra-carinal lymph nodes (white arrows). (Right): After 6 cycles of PRRT, follow-up somatostatin receptor imaging (including MIP and trans-axial PET/CT) showed a partial response with the reduction of lymph node manifestations, along with a substantial downregulation of in vivo SSTR expression.

**Table 1 cancers-15-03595-t001:** Patient characteristics.

	Total Cohort (n = 67)	Treatment Cohort (n = 14)
**Demographic data**		
Age (years, mean, range)	61.2 (41–80)	65.3 (48–79)
Female	27 (40.3%)	3 (21.4%)
**Previous therapies**		
Surgery	7 (10.4%)	2 (14.2%)
Radiation treatment	48 (71.6%)	8 (57.1%)
1st line CTx (cis-/carboplatin, etoposide)	66 (98.5%)	14 (100.0%)
2nd line CTx (topotecan, gemcitabine or ixoten)	30 (46.3%)	11 (78.6%)
Other systemic treatment *	2 (3.0%)	0
**Metastases**		
Lymph node	61 (91.0%)	14 (100.0%)
Bone	35 (52.2%)	8 (57.1%)
Liver	33 (49.3%)	6 (42.9%)
CNS	23 (34.3%)	5 (35.7%)
Adrenal	13 (19.4%)	3 (21.4%)
Lung	13 (19.4%)	3 (21.4%)
Pleura	11 (16.4%)	5 (35.7%)
Pancreas	4 (6.0%)	1 (7.1%)
Other Locations	9 (13.4%)	1 (7.1%)

* Other systemic treatments include everolimus and/or nivolumab/ipilimumab.

**Table 2 cancers-15-03595-t002:** Detailed characteristics of patients that underwent PRRT.

Sex	Age	Previous Therapies	Sites of Disease	SSTR Positivity	Number of PRRT Cycles	Cumulative Activity (GBq)	Best Response	PFS	OS
F	74	Surgery (primary), adjuvant RCTx (1st line), CTx (1st/ 2nd line)	Lymph nodes, pleura	all	5	39.5	PR	766	907
M	69	CTx (1st/ 2nd line)	Lymph nodes, skeletal, adrenal, pulmonary, pleura	all	1	7.4	SD	118	118
M	66	CTx (1st/ 2nd/ 3rd line)	Lymph nodes, skeletal, pleura	all	1	7.6	PD	53	53
M	48	CTx (1st/ 2nd line)	Lymph nodes, cerebral, pulmonary	majority	6	44.3	SD	396	575
M	55	RCTx (1st line), CTx (1st line), RTx (prophyl. brain, metastasis), CTx (2nd line), RCTx (metastasis)	Lymph nodes, skeletal, cerebral	all	2	14.8	PD	94	94
M	69	RCTx (1st line), Surgery (metastasis), RTx (metastasis), CTx (1st line)	Lymph nodes, skeletal, hepatic, cerebral	majority	1	6.0	PD	53	187
M	68	CTx (1st/2nd line)	Lymph nodes, hepatic, cerebral, adrenal, pleural	majority	1	6.9	PD	70	134
M	79	CTx (1st/2nd line)	Lymph nodes, skeletal, hepatic	majority	1	7.21	PD	56	56
M	70	RCTx (1st line)	Lymph nodes	all	3	21.3	SD	357	480
F	67	CTx (1st/2nd line)	Lymph nodes, skeletal, hepatic	all	1	7.3	PD	118	118
F	59	CTx (1st line), RTx (metastasis), CTx (2nd line)	Lymph nodes, pulmonary	all	2	15.2	PD	114	216
M	61	CTx (1st line), RTx (metastasis), CTx (2nd line)	Lymph nodes, adrenal, pancreatic, renal	majority	1	8.0	PD	103	103
M	70	CTx (1st line), RTx (residual tumor)	Lymph nodes, skeletal, hepatic, cerebral, pleural	majority	1	7.7	N/A *	17	17
M	59	CTx (1st line), RTx (prophyl. brain), CTx (2nd line)	Lymph nodes, skeletal, hepatic	majority	3	22.6	SD	132	198

SSTR, somatostatin receptor. PRRT, peptide receptor radionuclide therapy. GBq, Gigabecquerel. PFS, progression-free survival. OS, overall survival. RTCx, radiation and chemotherapy. CTx, chemotherapy. PR, partial response. PD, progressive disease. SD, stable disease. N/A, not available. * succumbed to treatment-independent infectious complications immediately after PRRT.

## Data Availability

Due to the European regulations regarding data protection, we cannot make data available online or disburse them. However, all data are available for revision on-site for the appropriate reason.

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
