# Peer review of "In-Vivo Somatostatin-Receptor Expression in Small Cell Lung Cancer as a Prognostic Image Biomarker and Therapeutic Target"

_cancers, 2023, doi:10.3390/cancers15143595_

Round 1
Reviewer 1 Report
Overall a very interesting work showing that an existing radiopharmaceutical therapy can be applied to additional diagnoses. Some remarks:
Was any FDG imaging done, how does that relate to the SSTR PET results?
Please improve table 1 layout under previous therapies to make it easier to see what number corresponds to what therapy.
Page 3 Scan interpretation section - Please provide further data on what "workstation" was used for evaluation of scans. If possible with software revision numbers etc.
Page 6 PRRT result section - Perhaps explain further about why there was such a difference in the number of treatment cycles, was it due to observed response, adverse effects etc?
Page 6 PRRT result section - Did the pre-treatment SUVmax or T/L values correlate in any way with outcome after PRRT?
Page 6 line 210 - Since patient immune response is reduced by haemotoxicity of PRRT, how do you motivate that the inability to survive the pneumonia is "PRRT-unrelated"?
Reviewer 2 Report
Good paper, solid data and well presented.
I suggest to use as acronym for PET/CT with somatostatin analogues "SRS" instead of PET, which is usually used for FDG.
My main concern is about FDG PET/CT.
Did patients performed also an FDG PET? If not, why not?
See also Martucci F, et al. Impact of 18F-FDG PET/CT in Staging Patients With Small Cell Lung Cancer: A Systematic Review and Meta-Analysis. Front Med (Lausanne). 2020 Jan 29;6:336. doi: 10.3389/fmed.2019.00336.
